# An Optimized SIFT-OCT Algorithm for Stitching Aerial Images of a Loblolly Pine Plantation

Tao Wu [1,2,3], I-Kuai Hung [4], Hao Xu [5], Laibang Yang [6], Yongzhong Wang [6], Luming Fang [1,2,3] and Xiongwei Lou [1,2,3,*]

1   College of Mathematics and Computer Science, Zhejiang A&F University, Hangzhou 311300, China
2   Key Laboratory of State Forestry and Grassland Administration on Forestry Sensing Technology and Intelligent Equipment, Zhejiang A&F University, Hangzhou 311300, China
3   Key Laboratory of Forestry Intelligent Monitoring and Information Technology Research of Zhejiang Province, Zhejiang A & F University, Hangzhou 311300, China
4   College of Forestry and Agriculture, Stephen F. Austin State University, Nacogdoches, TX 75962, USA
5   Zhejiang Forestry Bureau, Hangzhou 310000, China
6   Hangzhou Ganzhi Technology Co., Ltd., Hangzhou 310000, China
*   Correspondence: lxw@zafu.edu.cn; Tel.: +86-135-8822-8755

**Abstract:** When producing orthomosaic from aerial images of a forested area, challenges arise when the forest canopy is closed, and tie points are hard to find between images. The recent development in deep leaning has shed some light in tackling this problem with an algorithm that examines each image pixel-by-pixel. The scale-invariant feature transform (SIFT) algorithm and its many variants are widely used in feature-based image stitching, which is ideal for orthomosaic production. However, although feature-based image registration can find many feature points in forest image stitching, the similarity between images is too high, resulting in a low correct matching rate and long splicing time. To counter this problem by considering the characteristics of forest images, the inverse cosine function ratio of the unit vector dot product (arccos) is introduced into the SIFT-OCT (SIFT skipping the first scale-space octave) algorithm to overcome the shortfalls of too long a matching time caused by too many feature points for matching. Then, the fast sample consensus (FSC) algorithm was introduced to realize the deletion of mismatched point pairs and improve the matching accuracy. This optimized method was tested on three sets of forest images, representing the forest core, edge, and road areas of a loblolly pine plantation. The same process was repeated by using the regular SIFT and SIFT-OCT algorithms for comparison. The results showed the optimized SIFT-OCT algorithm not only greatly reduced the splicing time, but also increased the correct matching rate.

**Keywords:** feature matching; forest image stitching; SIFT-OCT; FSC

## 1. Introduction

In the biosphere, the forest not only has irreplaceable economic benefits for human beings but also has the ecological benefits of maintaining the balance of the terrestrial ecosystem [1,2]. Forest inventory helps to timely grasp the quantity and quality of forest resources, understand the dynamic rules of production and extinction, explore the relationship between the natural environment and economy, formulate and adjust forestry policies, and develop forest plans, so as to ensure that forest resources are fully utilized and maintained in national economic construction [3,4]. With the development of computer-related technologies, the application of deep learning technology to forest resource assessment has become a research hotspot [5,6]. Çalişkan et al. [7] used three network models, i.e., ResNet-18, MobileNet-V2, and Xception, to conduct the extraction of forest roads from high-resolution orthomosaic images. Lou et al. [8] applied three object detection algorithm models, i.e., Faster-RCNN, YOLO v3, and SSD, onto high-resolution orthomosaic images to

measure the tree crown size of young and mature loblolly pine stands. In Jie et al. [9], multiple high-resolution orthomosaic images were with three models: Faster-RCNN, FPN, and SSD, to detect pine wilt disease. The prerequisite for these image processing applications is the acquisition of high-quality orthomosaic images. The acquisition of orthophotos over forested areas, especially UAV-based high-precision orthophotos, is relatively difficult, due to the severe homogenization of the forest structure [10].

The core technology of orthomosaic image generation is image stitching, which is the process of registering two or more images of the same scene at different times with different sensors and viewpoints [11,12]. Image stitching technology can be divided into two categories, grey value extraction algorithms and feature extraction algorithms, based on the different methods of using image information [13]. Grey value extraction algorithms do not require feature extraction, but directly use the grey value information of the image for similarity measurement [14]. The commonly used grayscale-based methods are the normalized grey combination related law (NIC) and normalized product correlation matching algorithm (Nprod) [15]. However, for forest images collected by UAVs, they are often dominated by green color in leaf-on season. It was found through experiments with the grayscale algorithm that the grayscale values of image pixels were concentrated in a certain interval, due to the similarity in color and texture. Hence, when using the distance algorithm for matching grayscale images, it resulted in more false matching point pairs. Therefore, the matching algorithm, based on gray-scale correlation, is not suitable for stitching forest area images [16]. In contrast, feature-based matching algorithms detect corners, spots, lines, and other features found in images [17], of which the scale-invariant feature transform SIFT algorithm [18] is one of the most commonly used algorithms for image stitching. This algorithm maintains good robustness to image rotation, scaling, and translation, and has good processing ability for changes in illumination and the camera viewpoint. At present, academics have proposed several improved algorithms, based on the SIFT algorithm. Ke et al. [19] proposed the PCA-SIFT algorithm, which uses principal component analysis (PCA) to reduce the dimensionality of feature descriptors, resulting in an increase in the speed of feature point matching. Xiang et al. [20] proposed the OS-SIFT algorithm for optical image registration, which uses two Harris scale spaces for keypoint detection, direction assignment, descriptor extraction, and keypoint matching; the results showed that the method had more robust alignment for optical-to-SAR images and outperformed other algorithms, in terms of alignment accuracy. Ma et al. [21] introduced a new gradient definition to overcome image intensity differences between remote sensing image pairs, and an enhanced feature matching method was introduced to increase the number of correct correspondences by combining the position, ratio, and orientation of each key point. Their results showed that the method improved in the number of correct correspondences and alignment accuracy, compared with several existing methods. Ye et al. [22] used the combined features of CNN and SIFT that were incorporated into the PSO-SIFT algorithm for registration, which was superior, in terms of alignment accuracy and the number of correct correspondences. There are few studies on the stitching algorithm, aimed to process images of forested areas, where the number of extracted feature points is high, but the number of effective feature point pairs is low, leading to a long splicing time with low accuracy outcome at the same time. In this project, we proposed improving the image stitching process by optimizing the SIFT-OCT algorithm and realizing images of forest areas and assessed the outcomes, based on two statistics, i.e., the correct matching rate and stitching time.

## 2. Materials and Methods

### 2.1. SIFT-OCT Algorithm Description

The human eyes can distinguish objects in a certain range. However, if we want computers to do the same, computers need to have a unified understanding of objects at different scales; that is, to find out the features with scale invariance. The feature vector of the SIFT algorithm can keep invariance to rotation, scale, and brightness change. However,

due to the high dimension of SIFT feature vector, the matching operation of the feature vector is slow. Therefore, Schwind et al. [23]. proposed the SIFT-OCT algorithm, which skips the first set of scale-space octave for feature point detection on the basis of SIFT algorithm, so as to reduce the splicing time and improve the correct matching rate. The research shows that precision registration is related to the distribution properties and positional accuracy of the feature points. When extracting features, the SIFT-OCT algorithm can still maintain the subpixel accuracy of SIFT algorithm, without affecting the extraction accuracy of feature points. The feature points detected in large-scale space are more stable, which can remove the influence of fine, uneven texture on the images, so as to improve the correct matching rate.

The SIFT-OCT algorithm mainly includes four steps: (1) build scale space, (2) detect spatial extreme values, (3) locate feature points, and (4) generate feature vector.

(1) Scale-space construction is to identify potential key points by scanning images in position and proportion. Lindeberg's [24] study showed that Gaussian convolution was the only linear kernel function that could realize image scale transformation. Therefore, the construction of image scale space can be obtained by convolution of Gaussian function with an image. Gaussian convolution kernel is:

$$G(x,y,\sigma) = \frac{1}{2\pi\sigma^2}e^{(-\frac{x^2+y^2}{2g^2})} \tag{1}$$

Gaussian differential scale space is:

$$D(x,y,\sigma) = (G(x,y,k\sigma) - G(x,y,\sigma)) \times I(x,y) = L(x,y,k\sigma) - L(x,y,\sigma) \tag{2}$$

where $L(x,y,\sigma)$ is the scale space, $G(x,y,\sigma)$ is the Gaussian convolution kernel, $I(x,y)$ represents an image, and σ is the scale factor, also known as the Gaussian convolution smoothing factor.

(2) The spatial polar point detection is required to detect the candidate feature points after constructing the differential scale space. The SIFT-OCT algorithm starts the spatial polar search from the second set of differential scale space. It compares the pixel point, with the 26-pixel points in the upper and lower scales and $3 \times 3$ matrix of the scale, where the pixel point is located, and if the grey value of the point is maximum or minimum, then the point is marked as a candidate feature point.

(3) After the feature points are detected, it is necessary to accurately locate the specific location of each feature point. The main direction of the feature point is obtained, and the gradient distribution characteristics of the pixels in the domain of the feature point are used to determine its orientation parameters. Then, the gradient histogram of the image is used to obtain the stable direction of the local structure of the feature point. The gradient size is:

$$m(x.y) = \sqrt{[L(x+1,y) - L(x-1,y)]^2 + [L(x,y+1) - L(x,y-1)]^2} \tag{3}$$

The direction is:

$$\theta(x,y) = tan^{-1}\{[L(x,y+1) - L(x,y-1)]/[L(x+1,y) - L(x-1,y)]\} \tag{4}$$

(4) After accurately locating the feature points, one or more descriptors need to be established for each feature point, so that the descriptors have good invariance to scale, rotation changes, illumination changes, and perspective changes of the image. As shown in Figure 1, an $8 \times 8$ equal square window is constructed around the feature point, and its gradient value is calculated for each pixel in the window. Then, the $2 \times 2$ equal square window on the right is obtained by merging the calculations. Each direction after merging has eight directional values, so as to determine the 32-dimensional descriptor of the feature point. According to the suggestion made by Lowe [25], in the specific merging calculation process, a $4 \times 4$ equidistant square window can also be used to construct a 128-dimensional

vector to describe the central pixel, and the stability of matching will be stronger, where the matching of feature points is mainly achieved by the Euclidean distance.

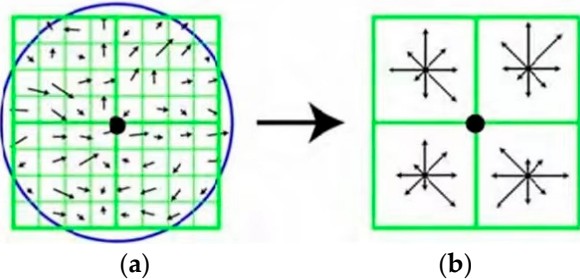

**Figure 1.** The feature vector of the SIFT-OCT algorithm vector. (**a**) Neighborhood gradient direction. (**b**) Key point eigenvectors.

### 2.2. Improved SIFT-OCT Algorithms

Currently, a SIFT-OCT based image stitching algorithm first detects and describes SIFT-OCT feature points in differential scale space. Then, it uses Euclidean distance to judge whether the feature points match, then optimizes and filters the correct matching pairs based on random sample consensus (RANSAC) algorithm [26]; finally, it performs image fusion to achieve image stitching. However, images of forested areas possess some challenges in this image stitching process. (1) Because the SIFT-OCT algorithm detects a large number of feature points in a forest area image and the corresponding feature descriptor dimensions are too high, it leads to too long of splicing time during the process of feature point matching. (2) Because of the single color, no intuitive outline, and low contrast of a forest image, the SIFT-OCT algorithm detects more feature points; however, after filtering and purification, the correct matching pairs are still low. The RANSAC algorithm is iteratively computed and filtered within the set of SIFT-OCT feature points. In order to achieve higher accuracy, the size of the set of matching points cannot be too small. Therefore, the RANSAC algorithm is not effective in stitching forest images.

In our project, the SIFT-OCT algorithm was applied to forest area images. In order to shorten the feature matching time and reduce the computational complexity, arccos was used to replace the Euclidean distance at this feature point matching stage. Next, the fast sample consensus (FSC) algorithm [27] was introduced to replace the RANSAC algorithm at the purification and optimization stage on the feature point matching point pairs, in order to remove the mismatched point pairs, improve the correct rate, and achieve a more appropriate number of feature points and their distribution. Thus, the stitching time can be greatly reduced, correct matching rate can be improved, and forest images can be stitched simultaneously.

#### 2.2.1. Feature Point Matching Strategy Optimization

The SIFT-OCT algorithm uses the Euclidean distance ratio to determine whether feature points match. For feature descriptors in reference images $e_l$, it finds the distance between $e_l$ and the next closet feature descriptors, $e_r$ and $e_q$, in the image to be aligned. Then, the ratio $N$ of Euclidean distance $D(e_l, e_r)$ to $D(e_l, e_q)$ is calculated.

$$N = \frac{D(e_l, e_r)}{D(e_l, e_q)} = \frac{\sqrt{\sum_{i=1}^{128}(e_{li} - e_{ri})^2}}{\sqrt{\sum_{i=1}^{128}(e_{li} - e_{qi})^2}} \tag{5}$$

In the equation above, $e_l = (e_{11}, e_{l2} \dots e_{l28})$, $e_r = (e_{r1}, e_{r2} \dots e_{r128})$, $e_q = (e_{q1}, e_{q2} \dots e_{q128})$. In application, a radio threshold is set as M. If N < M, it keeps the $(e_l, e_r)$ pairs of feature points as a matching point pair; otherwise, it is discarded. Following such a matching method can find suitable matching pairs, but the computation process is more complicated, resulting in a higher time cost.

In order to simplify the matching process and improve the speed of feature point matching, this project introduced arccos of unit vector for matching decision, instead of Euclidean distance. The feature points in one image are dotted with all the feature points in the other image, and the inverse cosine is calculated to obtain the angle set. The minimum angle $\theta_1$ and next smallest angle $\theta_2$ are found from the angle set. If the ratio of the two is less than a specified radio M, the feature point corresponding to the minimum angle is considered to be successfully matched with the feature point in the other image.

$$\frac{\theta_1}{\theta_2} = \frac{arccos(e_l e_r)}{arccos(e_l e_q)} = \frac{arccos \sum_{i=1}^{128}(e_{li}e_{ri})}{arccos \sum_{i=1}^{128}(e_{li}e_{qi})} \tag{6}$$

As can be seen from the above equations, square and root sign operations are needed several times when using Euclidean distance for matching calculation, which is a tedious calculation process with low matching efficiency. In contrast, the calculation method adopted in this project only requires basic operations, such as vector multiplication and inverse cosine function, which greatly simplifies the calculation process and effectively improves the efficiency of feature point matching. In this project, we calculated the distance by Euclidean distance and arccos for 10,000 randomly generated data of 128 dimensions, and measured the time required for processing each of the two distance equations used. On the computer with the same configuration, the time required for Euclidean distance was 0.3065 s, compared with 0.1333 s for arccos. The time required for arccos was only 43.5% of Euclidean distance, which proved that the arccos is significantly more efficient in calculating the similarity of two feature point matching.

### 2.2.2. Feature Point Matching Pair Strategy Optimization

After obtaining the matched pairs of the feature points, a large number of outliers may exist. Therefore, the matched pairs need to be purified and optimized to obtain the optimal image transformation matrix for image stitching. Many methods use the RANSAC algorithm to obtain robust results. However, this algorithm is a random sampling consistency algorithm. The principle is to estimate the model parameters by randomly selecting a certain number of samples and calculating the coordinate transformation relation between the feature points of the reference image and corresponding feature points of the image to be matched. The RANSAC algorithm eliminates mismatched points and calculates the errors of matched points after positive and inverse transformations of the transformation matrix. By using the ratio set, the points with larger errors are eliminated, and an optimized pair of correct matched points is obtained. Especially when the authority interior-point ratio is less than 50%, the results of the RANSAC algorithm are not ideal.

In contrast, the FSC algorithm improves the reliability and efficiency of the algorithm by obtaining a subset with a high matching rate from the set of matched point pairs and then sampling within that subset to obtain the maximum consistent set. The FSC algorithm first requires a set of observations as input data and then selects a parametric model for this set of observations and set of parameters with high confidence for the model. The input data are distinguished into intra- and extra-local points, and the most appropriate model is computed by iteratively selecting a set of random subsets of the data. The specific process is:

1. First, a suitable model is chosen for the local points, and all unknown parameters of the model are obtained by calculation.
2. Second, the model is used to test outlier points, and if the data for an outlier point also applies to the model, then that outlier point will also be converted to an inlier point.
3. By analogy, if a sufficient number of extrinsic points are converted to intrinsic points, the model is deemed appropriate.
4. Finally, estimation and error analysis of the model using all intra-local points to assess the accuracy of the model.
5. The above process is repeated n times, and the model with a higher number of points in the bureau, and a higher accuracy rate is selected as the best model.

From the principle of feature point matching in the SIFT-OCT algorithm, it is known that the threshold value of the similarity measure ratio at matching affects the number of matched points and correctly matched pairs. In the FSC Algorithm 1, the corresponding SIFT-OCT feature point sets $C_h$ and $C$ are first matched according to two thresholds, one large and one small. Then, $C_i^h$, $C_j^h$, and $C_k^h$ are randomly selected from $C_h$ with high correctness to calculate the corresponding transformation parameter, and the transformation error with each point pair in the point set $C$ is calculated using the transformation parameter; the point pair with less than one pixel error is added to $C_i$, and the corresponding point pair in $C_i$ is used to calculate the transformation parameter again. This process is repeated a fixed number of times to determine the optimal distance ratio.

---

**Algorithm 1** Fast Sample Consensus (FSC)

---

**Input:**

- $C_h$: the sample correspondence set.
- $C$: the total tentative correspondence set.
- $N$: number of iterations.

**Output:** the transformation model parameters

1　　n = 0.
2　　**for** $i = 1: N$.
3　　Randomly select three correspondences $C_i^h$, $C_j^h$, and $C_k^h$ from $C^h$ .
4　　Calculate the transformation model parameters $\theta_i$ by correspondences $C_i^h$, $C_j^h$, and $C_k^h$ .
5　　Calculate the transformation error of every correspondence in the set $C$ by model parameters $\theta_i$, and consensus set $C_i$ is made up of the correspondences with error less than 1 pixel.
6　　**if** $size(C_i) > $ n, **do**
7　　n= $size(C_i)$
8　　Calculate
9　　**end if**
10　　**end for**

---

2.2.3. Assessment Criteria

To assess the performance of an algorithm used for stitching images of a forested area, the following two statistics were used.

(1) Correct matching rate: The correct matching rate is the ratio of the number of correctly matched feature points in the image matching to the total number of feature matching. Under different calculation principles, the meanings of the two numbers are also different. The correct rate can reflect the matching effect under certain constraints.

$$Accuracy = \frac{Correct\ point\ logarithm}{Matched\ point\ logarithm}\% \qquad (7)$$

(2) Stitching time: The image stitching time reflects the real-time performance of the stitching algorithm, in which the algorithm is executed 10 times, and the average running time of the 10 times is used as the final stitching time of the algorithm.

2.2.4. Materials

The study site is a loblolly pine plantation (*Pinus taeda*) located in Cherokee County ($31°45′31.3″$ N, $95°02′31.8″$ W) of east Texas, USA. It was converted from an old field in 2001 for timber production. The pine seedlings were initially planted in rows. Some thinning treatments have been applied recently. A DJI Phantom 4 Pro V2.0 UAV was used to capture aerial images of the study area. The UAV was flown at an altitude of 40 m above ground. The course overlap was 90%, and the side overlap 90%, as well. The images had a dimension of 5472 × 3648 pixels. Three sets of images representing different ground covers were selected for the image stitching process. As shown in Figure 2, images a1 and

a2 represent an area located at the center of the forest, while b1 and b2 showed the area along the edge of the forest; c1 and c2 covered a forest road.

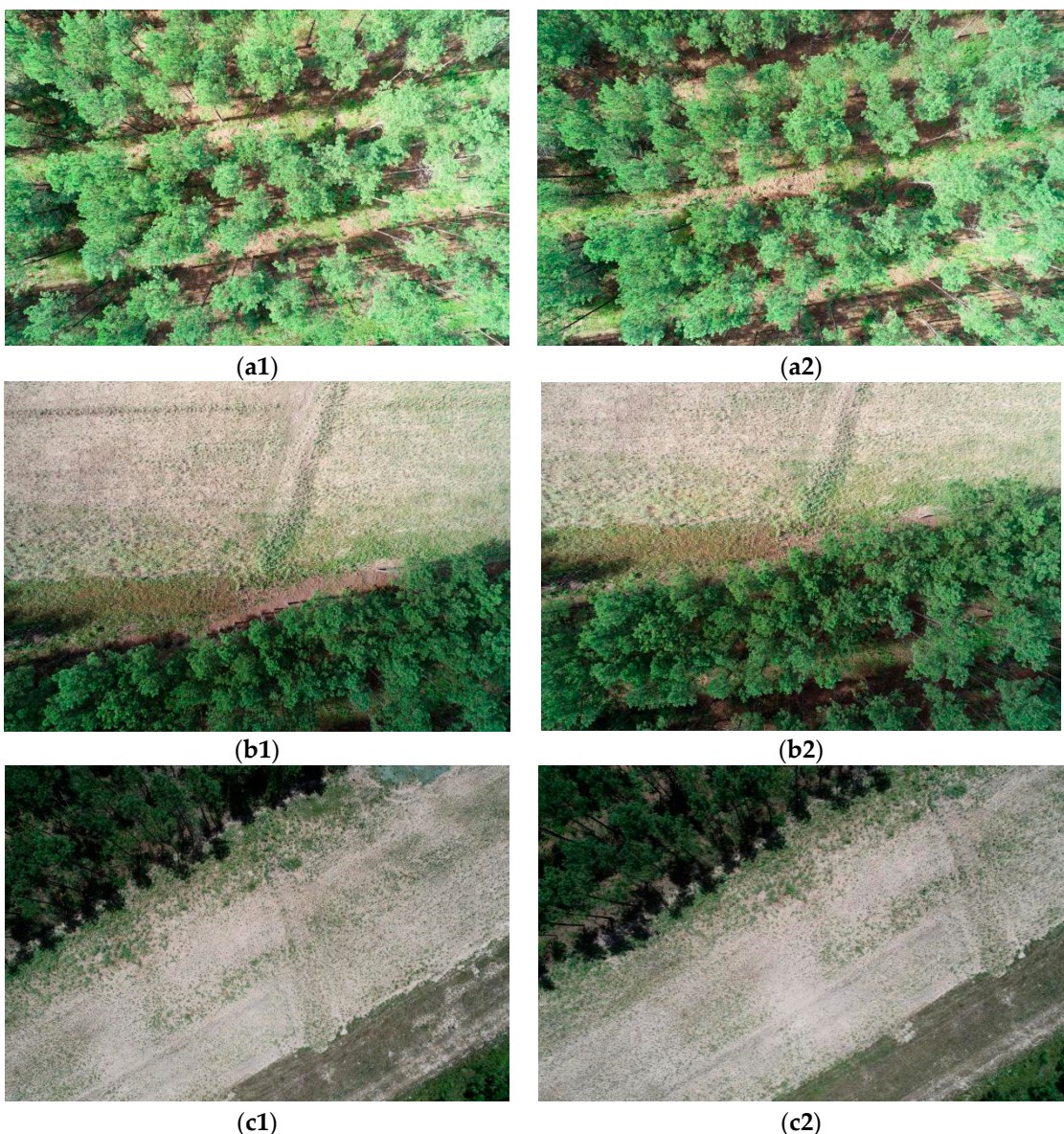

(**a1**)  (**a2**)

(**b1**)  (**b2**)

(**c1**)  (**c2**)

**Figure 2.** Three image pairs represent the forest core, forest edge, and forest road. (**a1**) Forest core left image. (**a2**) Forest core right image. (**b1**) Forest edge left image. (**b2**) Forest edge right image. (**c1**) Forest road left image. (**c2**) Forest road right image.

## 3. Results and Discussion

In order to determine the performance of our improved SIFT-OCT algorithm, the image dataset was processed for image stitching. The outcome was compared to those processed using the SIFT and original SIFT-OCT algorithms. These algorithms are based on MATLAB R2018a software environment. The computer used for data processing had an Intel(R) Xeon(R) Silver 4110 CPU with a clock rate of 2.10 GHz, 64 GB RAM, and a NVIDIA GeForce GTX 1080 Ti graphic processor with 11 GB memory. The statistics of each algorithm were recorded for comparison. Of those, the splicing time and correct matching rate, presented as percent accuracy, were used as the evaluation criteria for algorithm performance comparison. The results of the three sets of images by three different algorithms are shown in Table 1. The results of feature matching and splicing effects on

linear features of the three sets of images by three different algorithms are shown in Figures 3–5.

**Table 1.** Comparison of image stitching algorithm efficacy.

| Image Pair | Algorithm | Number of Feature Points Left | Right | Number of Matched Points | Number of Correct Points | Accuracy (%) | Splicing Time (s) |
|---|---|---|---|---|---|---|---|
| Core a1/a2 | SIFT | 92,928 | 89,250 | 1764 | 868 | 49.21 | 1074.11 |
| | SIFT-OCT | 15,689 | 15,346 | 147 | 74 | 50.34 | 308.08 |
| | Optimized SIFT-OCT | 15,689 | 15,346 | 148 | 83 | 56.08 | 74.42 |
| Edge b1/b2 | SIFT | 84,521 | 86,386 | 18,444 | 9114 | 49.43 | 935.22 |
| | SIFT-OCT | 9212 | 8370 | 1619 | 825 | 50.96 | 148.96 |
| | Optimized SIFT-OCT | 9212 | 8370 | 1628 | 1149 | 70.58 | 70.17 |
| Road c1/c2 | SIFT | 81,054 | 80,193 | 7228 | 3231 | 44.70 | 858.17 |
| | SIFT-OCT | 7307 | 7477 | 1313 | 564 | 42.96 | 110.39 |
| | Optimized SIFT-OCT | 7307 | 7477 | 1326 | 683 | 51.51 | 57.66 |

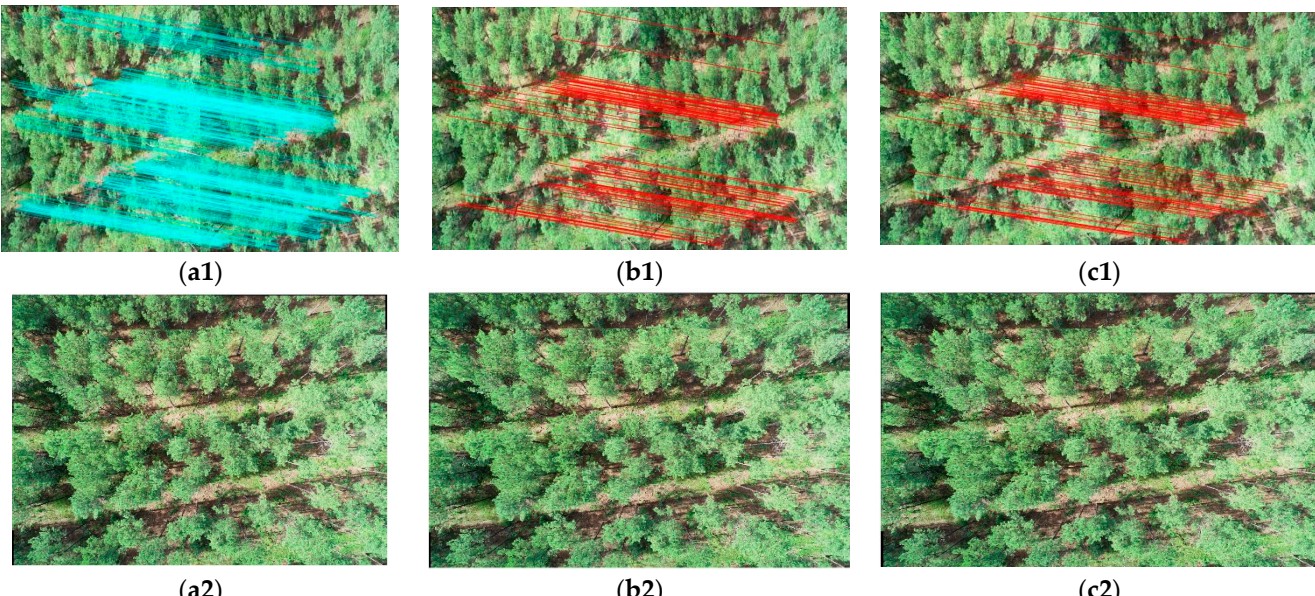

(**a1**)  (**b1**)  (**c1**)

(**a2**)  (**b2**)  (**c2**)

**Figure 3.** Feature matching and splicing effects on linear features in the forest core area. (**a1**) Feature matching results of the SIFT algorithm. (**a2**) Splicing effect of the SIFT algorithm. (**b1**) Feature matching results of the SIFT-OCT algorithm. (**b2**) Splicing effect of the SIFT-OCT algorithm. (**c1**) Feature matching results of the optimized SIFT-OCT algorithm. (**c2**) Splicing effect of the optimized SIFT-OCT algorithm.

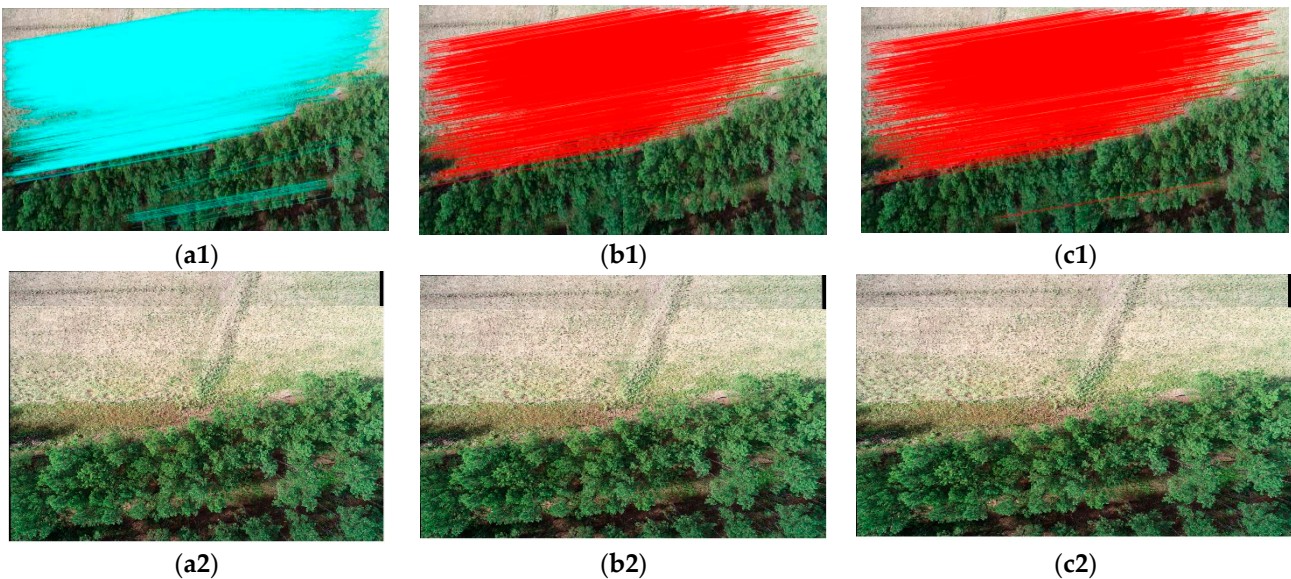

**Figure 4.** Feature matching and splicing effects on linear features in the forest edge area. (**a1**) Feature matching results of the SIFT algorithm. (**a2**) Splicing effect of the SIFT algorithm. (**b1**) Feature matching results of the SIFT-OCT algorithm. (**b2**) Splicing effect of the SIFT-OCT algorithm. (**c1**) Feature matching results of the optimized SIFT-OCT algorithm. (**c2**) Splicing effect of the optimized SIFT-OCT algorithm.

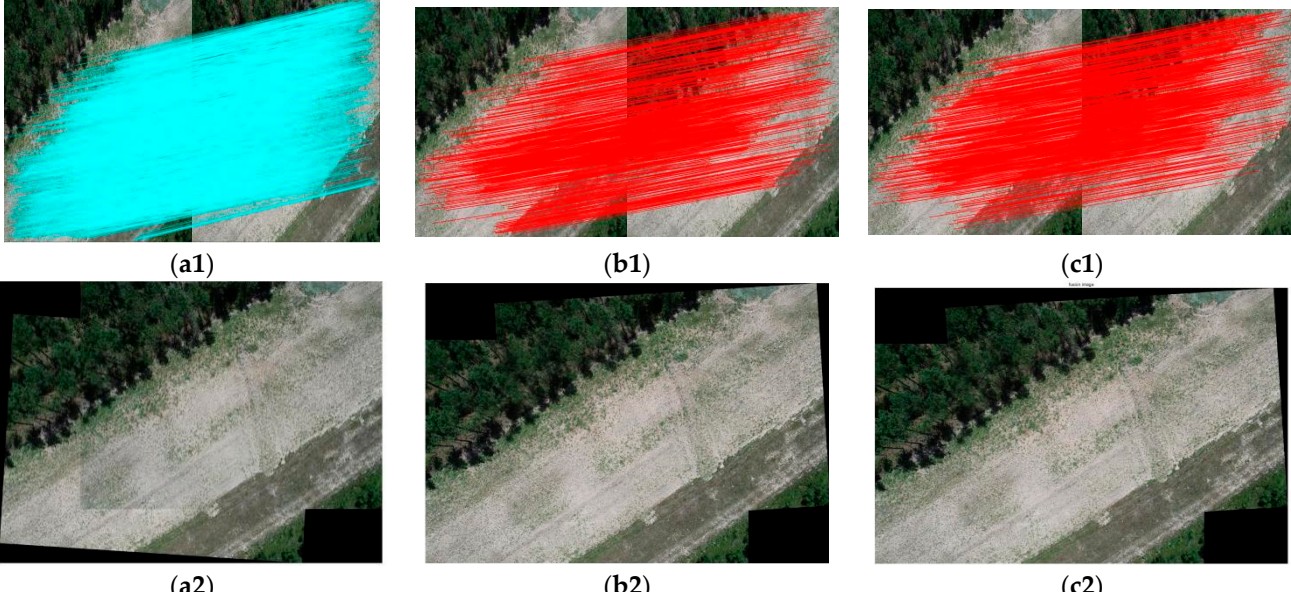

**Figure 5.** Feature matching and splicing effects on linear features in the forest road area. (**a1**) Feature matching results of the SIFT algorithm. (**a2**) Splicing effect of the SIFT algorithm. (**b1**) Feature matching results of the SIFT-OCT algorithm. (**b2**) Splicing effect of the SIFT-OCT algorithm. (**c1**) Feature matching results of the optimized SIFT-OCT algorithm. (**c2**)Splicing effect of the optimized SIFT-OCT algorithm.

As seen in Figure 6, when stitching forest core, edge, and road images, the correct matching rates of SIFT algorithm were 49.21%, 49.43%, and 44.70%, respectively, while the SIFT-OCT algorithm resulted in 50.43%, 50.96%, and 42.96%. These two commonly used algorithms achieved about the same level of accuracy. In contrast, the performance of the optimized SIFT-OCT algorithm achieved higher accuracy than the two other algorithms in all of the three ground cover categories, i.e., center, edge, and road, with the correct match-

ing rates of 56.08%, 70.58%, and 51.51%, respectively. This increase in matching accuracy is accomplished by introducing the FSC algorithm to replace the RANSAC algorithm in the feature point purification and optimization stage. The performance of the optimized SIFT-OCT algorithm is particularly outstanding in matching forest edge images, with its correct matching rate being as high as 70.58%. Compared with 49.43% of SIFT algorithm and 50.96% of the SIFT-OCT algorithm in matching forest edge images, the difference of 21.15% and 19.62% is a big improvement.

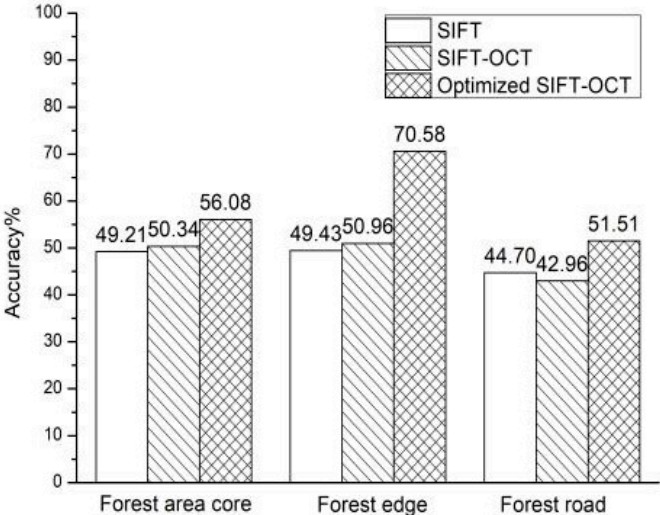

**Figure 6.** Matching accuracy comparison between different algorithms on different ground covers.

The time consumption of the stitching process mainly focuses on four aspects: feature point extraction, feature point description, feature point matching, and image fusion. Among them, it takes a long time in the stage of feature point description and point matching. The SIFT algorithm takes the longest time to stitch forest core (1074.11 s), edge (935.22 s), and road (858.17 s) images. In contrast, since the SIFT-OCT algorithm actively skipped the first set of scale-space for feature point detection in the feature point extraction stage, the number of feature points detected in the feature point extraction stage was greatly reduced. Therefore, in the feature point matching stage, due to the reduction of the number of feature points, the time required for stitching was also greatly reduced, consuming only 308.08 s (center), 148.96 s (edge), and 110.39 s (road) in stitching the three categories of images. Compared with the other two algorithms, the optimized SIFT-OCT algorithm introduced the ratio of the inverse cosine function of the unit vector point product to replace the Euclidean distance in the feature point matching stage, simplified the calculation formula, effectively improved the feature point matching efficiency, and further improved the splicing efficiency. The time required for these optimized algorithms was only 74.42 s (center), 70.17 s (edge), and 57.66 s (road). Among them, the optimized SIFT-OCT algorithm is particularly prominent in matching the images of the forest core area. It consumed only 74.42 s, much lower than that of the SIFT (1074.11 s) and SIFT-OCT (308.08 s) algorithms. In comparison, it was only 6.93% and 24.16% of the time required for stitching, respectively.

## 4. Conclusions

In this project, we improved the SIFT-OCT algorithm based on forest area image features and realize the stitching of forest images by introducing arccos and FSC algorithm. For comparison, three algorithms, i.e., the SIFT, SIFT-OCT, and optimized SIFT-OCT algorithms, were used to splice the forest core, edge, and road area images, respectively. The experimental analysis was conducted to assess the correct matching rate and splicing time. The results showed that all three algorithms were capable of stitching forest images, with around 50% accuracy of correct feature matching. Among them, the optimized SIFT-OCT

algorithm performed best in both the correct matching rate and stitching time. The correct matching rate was much higher than others in matching forest edge images. At the same time, the required stitching consumption time for the optimized SIFT-OCT was significantly reduced, compared with the SIFT and SIFT-OCT algorithms. This process of time reduction is of importance when processing a larger number of images. The optimized SIFT-OCT algorithm has good robustness and adaptability to realize the stitching of images of different forest types with rapid alignment and stitching of high-resolution aerial images. It also leads to the production of a high-quality forestry orthophoto mosaic in real-time, which allows for using deep learning for tree crown identification and timber volume estimation. The applicable scenario of this optimized algorithm is focused on the stitching of high resolution forest images, which requires a large number of feature points and matching pairs. When applying to other types of imagery other than forest, the number of detected feature points and correct matching pairs might be small, which would show less advantage for using this algorithm. Given access to other types of images, for example, different forest types and different land cover types, this algorithm can be tested on a variety of scenarios.

**Author Contributions:** Conceptualization, X.L.; formal analysis, X.L., Y.W., and L.Y.; funding acquisition, L.F.; methodology, H.X.; resources, X.L. and I.-K.H.; writing-original draft, T.W. All authors have read and agreed to the published version of the manuscript.

**Funding:** This research received no external funding.

**Institutional Review Board Statement:** Not applicable.

**Informed Consent Statement:** Not applicable.

**Data Availability Statement:** No new data were created or analyzed in this study. Data sharing is not applicable to this article.

**Conflicts of Interest:** The authors declare no conflict of interest.

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
