# Peer review of "An Optimized SIFT-OCT Algorithm for Stitching Aerial Images of a Loblolly Pine Plantation"

_forests, doi:10.3390/f13091475_

Round 1
Reviewer 1 Report
Nice work with all elements for scientific research. Results are vell presented. I can recommend publication for article as is. I don't feel qualified to judge about the English language and style
Reviewer 2 Report
1. Dear Authors,
I appreciate both the scientific and practical value of the paper, as well as your systematic work on it. Thank you for this valuable contribution along with detailed explanations. I am satisfied, knowing how complicated is the subject you dealt with when going into details.
I would like to request a minor revision related to use of multiple acronyms in the text; for example, please quote "fast sample consensus (FSC) at first mention in the text and subsequently use acronym. Accordingly, revise all the other acronyms as well or alternatively you can provide a full list with details prior to references.
Thank you again for this valuable contribution.
Reviewer 3 Report
Line 44 page 1
“Lou et al[8]. applied three objection detection algorithm models, Faster-RCNN, YOLO v3, and SSD, onto high-resolution orthmosaic images to measure the tree crown size of young and mature loblolly pine stands. “
Authors should clarify what is objection detection algorithm for the forest images collected by UAVs, the grayscale correlation-based matching algorithms are not suitable due to the single color, low contrast, high similarity between trees and relatively uniform tree canopy texture of forest images.
Strategy used by authors and their contribution
Refer to lines 157-165
“In our project, the SIFT-OCT algorithm was applied to forest area images. Firstly, the ratio of the arccosine of the unit vector dot product was used to replace the Euclidean distance in the feature point matching stage, which reduced the computational complexity and shortened the feature matching time. Secondly, the FSC algorithm[27] was introduced to replace the RANSAC algorithm in the phase of feature point matching, point pair purification, and optimization stage. This algorithm deleted mismatched point pairs, improved the correct matching rate, and resulted in a more appropriate number of feature points and their distribution. Thus, the stitching time can be greatly reduced, the correct matching rate can be improved, and the forest images can be stitched simultaneously”
Authors claim to improve the algorithm in terms of time and accuracy. This is achieved by (i) introduction of existing FSC in place of RANSAC (ii) use of arccos function instead of Euclidian distance. Interventions are good. The motivation for introducing these changes may be mentioned for more clarity.
Authors should explain these with the help of suitable small examples for more clarity and benefit to the readers
Refer to lines 176-189
In order to simplify the matching process and improve the speed of feature point matching, this project introduced the ratio of inverse cosine function of point product of unit vector for matching decision, instead of Euclidean distance. The feature points in one image are dotted with all the feature points in the other image, and the inverse cosine is calculated to obtain the angle set. The minimum angle θ1 and the next smallest angle θ2 are found from the angle set. If the ratio of the two is less than a specified radio M, the feature point corresponding to the minimum angle is considered to be successfully matched with the feature point in the other image….
As can be seen from the above equations, square and root sign operations are needed several times when using Euclidean distance for matching calculation, which is a tedious calculation process with low matching efficiency. In contrast, the calculation method adopted in this project only requires basic operations such as vector multiplication and inverse cosine function, which greatly simplifies the calculation process and effectively improves the efficiency of feature point matching.”
Authors may show how and why square and root operations are more time consuming as compare to arccos function for better clarity. The reference for the same or small suitable example may be helpful for more clarity.
It is commonly known that all algorithms are not suitable for all kinds of datasets. The existing algorithms (pre-improved versions) also may be showing improved accuracy in some other data sets. In this context, it is advised to use the proposed algorithm on some existing datasets of other researchers where the SIFT and SIFT-OCT have been used and compare the accuracy with the proposed algorithm. Besides, the authors should identify the limitation of the algorithm for the benefit of the readers.
There are repetitions in the article regarding use of arccos and FSC functions. Authors are advised to carefully examine and address them.
As the authors proposed an innovative methodology with good results, there needs to be clear presentaion of methodological aspects in terms of implementation e.g. algorithmic explanation, codes, libraries, software and limitation of the proposed algorithm. Besides, either it needs to be tested on other datasets already used by other benchmarking datasets or limitations must be clearly specified.
